# NLP-based Feature Extraction for the Detection of COVID-19 Misinformation Videos on YouTube

**Juan Carlos Medina Serrano, Orestis Papakyriakopoulos, Simon Hegelich**

Technical University of Munich, Germany

{juan.medina, orestis.p}@tum.de, simon.hegelich@hfp.tum.de

## Abstract

We present a simple NLP methodology for detecting COVID-19 misinformation videos on YouTube by leveraging user comments. We use transfer-learning pre-trained models to generate a multi-label classifier that can categorize conspiratorial content. We use the percentage of misinformation comments on each video as a new feature for video classification. We show that the inclusion of this feature in simple models yields an accuracy of up to 82.2%. Furthermore, we verify the significance of the feature by performing a Bayesian analysis. Finally, we show that adding the first hundred comments as tf-idf features increases the video classifier accuracy by up to 89.4%.

## 1 Introduction

The COVID-19 health crisis was accompanied by a misinfodemic: The limited knowledge on the nature and origin of the virus gave ample space for the emergence of conspiracy theories, which were diffused on YouTube, and online social networks. Although YouTube accelerated attempts to detect and filter related misinformation, it yielded moderate results (Li et al., 2020; Frenkel et al., 2020).

In this study, we present a simple NLP-based methodology that can support fact-checkers in detecting COVID-19 misinformation on YouTube. Instead of training models on the videos themselves and predicting their nature, we exploit the vast amount of available comments on each YouTube video and extract features that can be used in misinformation detection. Our methodology comes with the advantage that labeling comments is simpler and faster than video labeling. Additionally, no complex neural architecture is needed for the classification of videos.

Our study provides the following contributions:

- We create a multi-label classifier based on transfer-learning that can detect conspiracy-laden comments. We find that misinformation videos contain a significantly higher proportion of conspiratorial comments.

- Based on this information, we use the percentage of conspiracy comments as feature for the detection of COVID-19 misinformation videos. We verify its efficiency by deploying simple machine learning models for misinformation detection. We validate feature significance by Bayesian analysis.

- We show that including the first hundred comments as tf-idf features in the classifier increases the accuracy from 82.2% to 89.4%.

## 2 Related Work

Previous research studies have extensively investigated the possibilities and limits of NLP for detecting misinformation. Researchers have provided theoretical frameworks for understanding the lingual and contextual properties of various types of misinformation, such as rumors, false news, and propaganda (Li et al., 2019; Thorne and Vlachos, 2018; Rubin et al.; Zhou and Zafarani, 2018). Given the general difficulty in detecting misinformation, scientists have also developed dedicated benchmark datasets to evaluate the effectiveness of NLP architectures in misinformation related classification tasks (Pérez-Rosas et al., 2018; Hanselowski et al., 2018). Given the vast amount of misinformation appearing in online social networks, various research studies propose case-specific NLP methodologies for tracing misinformation. For example, Della Vedova et al. (2018) and Popat et al. (2018) combined lingual properties of articles and other meta-data for the detection of false news. Volkova et al. (2017), Qazvinian et al. (2011) and Kumar and Carley (2019) created special architectures that take into consideration the microblogging structure

of online social networks, while De Sarkar et al. (2018) and Gupta et al. (2019) exploited sentence-level semantics for misinformation detection.

Despite the deployment of such architectures for fact-checking, locating malicious content and promptly removing them remains an open challenge (Gillespie, 2018; Roberts, 2019). In the case of Covid-19 misinformation, a large share of conspiratorial contents remain online on YouTube and other platforms, influencing the public, despite content moderation practices (Li et al., 2020; Frenkel et al., 2020; Ferrara, 2020). Given this, it is important to develop case-specific NLP tools that can assist policymakers and researchers in the process of detecting COVID-19 misinformation and managing it accordingly. Towards this end, we illustrate how NLP-based feature extraction (Shu et al., 2017; Jiang et al., 2020; Lendvai and Reichel, 2016) based on user comments can be effectively used for this task. User comment data has been employed to annotate social media objects (Momeni et al., 2013), infer the political leaning of news articles (Park et al., 2011), and to predict popularity (Kim et al., 2016). Jiang and Wilson (2018) previously analyzed user comments to detect misinformation. However, they focused on linguistic signals and concluded that users' comments were not strong signals for detecting misinformation.

## 3 Methodology and Experiments

### 3.1 Dataset

The first step of the study consisted of obtaining a set of YouTube videos that included either misinformation or debunking content. We decided not to use YouTube's search function as previous studies found few conspiratorial content on the top results (Marchal et al., 2020). We preferred to search for YouTube videos through user-generated content on social media platforms. For this, we queried the pushhift Reddit API (Baumgartner et al., 2020), and Crowdtangle's historical data of public Facebook posts (Silverman, 2019) using the query "COVID-19 OR coronavirus". Additionally, we downloaded the COVID-19 Twitter dataset developed by Chen et al. (2020). The total dataset included over 85 million posts generated between January and April 2020. We significantly reduced this dataset by querying the posts with "biowarfare OR biological weapon OR bioweapon OR man-made OR human origin". From the remaining posts, we extracted and expanded the URLs. We

identified 1,672 unique YouTube videos. 10% of these videos had been blocked by YouTube as of April 2020. For the rest of the videos, we watched them, excluded the non-English videos, and manually labeled them as either misinformation, factual, or neither. To label a video as misinformation, we validated that its message was conveying with certainty a conspiracy theory regarding the origin of the coronavirus, as a man-made bioweapon or being caused by 5G. We did not classify videos that questioned its origin but showed no certainty about a hoax (which included well-known and verified news media videos) as misinformation. We classified as factual those videos that included debunking of conspiracy theories or presented scientific results on the origins and causes of COVID-19. We labeled the rest of the videos as neither. Two of the authors (JCMS, OP) performed the labeling procedure independently. For the cases where the labels did not agree, the third author was consulted (SH).

Afterward, we collected the comments on both misinformation and factual videos using YouTube's Data API[1]. For this study, we only included videos with more than twenty comments. The final dataset consisted of 113 misinformation and 67 factual videos, with 32,273 and 119,294 total comments respectively. We selected a ten percent random sample of the comments from the misinformation videos and proceeded to label them. This labeling procedure was performed in the same manner as the video classification to assure data quality. For each comment, we collected two labels. First, we gave a label if the comment expressed agreement (1) or not (0). Agreement comments included comments such as "this is the video I was looking for", or "save and share this video before YouTube puts it down". The second label considered if comments amplified misinformation with a conspiracy theory/misinformation comment (1) or without one (0). Comments that questioned the conspiracies (such as "could it be a bioweapon?") were not labeled as misinformation. 19.7% of the comments in the sample were labeled as conspiracy comment and 12.5% as agreement comment. Only 2.2% of the comments were classified as both agreement and conspiratorial. Although both agreement and conspiracy labeled comments express the same message of believing in the misinformation content from the videos, we decided to keep them apart due to their different linguistic properties. To com-

---

[1]https://developers.google.com/youtube/v3

pare the collection of agree-labeled comments and conspiracy-labeled comments, we tokenized and created a bag-of-words model. The two collections share 19.4% of their vocabulary. However, only 1.95% of the vocabulary has more than four occurrences in both collections. We applied $\chi^2$ tests for each of these remaining words and observe that 50% occur in significantly different proportions. At the end, only 0.96% of the vocabulary has a significant similar number of occurrences in the two datasets. The YouTube comments dataset without user data can be accessed in this GitHub repository[2], alongside a Google Colab notebook with the code.

### 3.2 Classification of Users Comments

We first performed a multi-label classification on the 10% sample of the misinformation videos' comments. We split the annotated data into training (80%) and test (20%) datasets. We employed state-of-the-art neural transfer-learning for the classification by fine-tuning three pre-trained models: XLNet base (Yang et al., 2019), BERT base (Devlin et al., 2018) and RoBERTa base (Liu et al., 2019). The fine-tuning consists of initializing the model's pre-trained weights and re-training on labeled data. We ran the models for four epochs using the same hyperparameters as the base models. For the experiments, we used 0.5 as a decision threshold. Additionally, we train two simpler models as baselines: a logistic regression model using LIWC's lexicon-derived frequencies (Tausczik and Pennebaker, 2010) as features, and a multinomial naive Bayes model using bag-of-words vectors as features. Table 1 shows the average micro-$F_1$ scores for the three transformer models after performing the fine-tuning five times. RoBERTa is the best performing model for the training and test dataset on the conspiracy classification as for

[2]https://github.com/JuanCarlosCSE/YouTube_misinfo

|  | Agree | | Conspiracy | |
|---|---|---|---|---|
|  | Train | Test | Train | Test |
| **LIWC** | 88.7 | 88.6 | 81 | 78.2 |
| **NB** | 94.2 | 82.4 | 94.3 | 78.8 |
| **XLNet** | 97±0.1 | 93.1±0.3 | 93.9±0.5 | 84.8±0.6 |
| **BERT** | **98.5**±0.1 | 93.3±0.5 | 96.3±0.3 | 83.8±0.9 |
| **RoBERTa** | 98.1±0.2 | **93.9**±0.4 | **96.4**±0.3 | **86.7**±0.5 |

Table 1: Train and test micro $F_1$ scores (mean and standard deviation) from multi-label classification models: LIWC with logistic regression and Naive Bayes as baselines, and three transformer models with five runs.

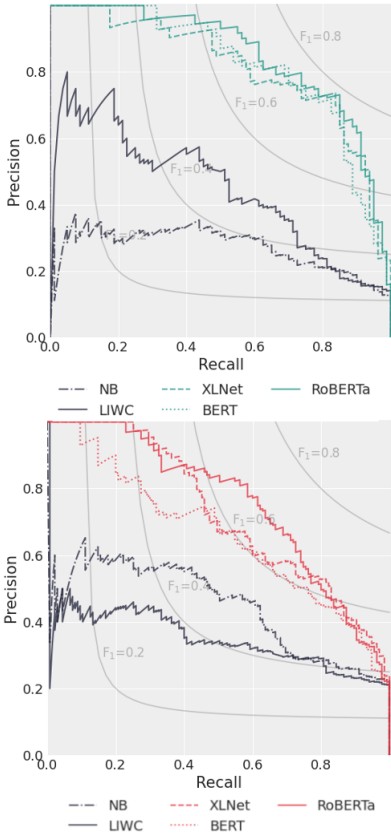

Figure 1: Precision and recall curves for binary $F_1$ scores for the conspiracy (upper figure) and agreement (lower figure) label. The plot shows the results for three neural-transfer classifiers.

the test data on the agreement label. BERT is the best performing model only for the training data on the agree label. The three transformer models outperform the baseline models. This predictive superiority is more evident in the precision-recall curves (with corresponding binary-$F_1$ scores) of the five models on the test data (Figure 1).

We employed the fine-tuned RoBERTa model to predict the labels of the remaining comments from the misinformation and factual videos. We then calculated the *percentage of conspiracy comments* per video. We also obtained this percentage for the agreement label. Figure 2 shows the resulting density distributions from misinformation and factual videos. We observe a difference between the distributions from the two types of videos. We confirmed this by performing Welch's t-test for independent samples. For the conspiracy comments percentage, the t-test was significant (p<0.000), indicating that the samples come from different distributions. The t-test was not significant for the agreement percentage (p>0.1).

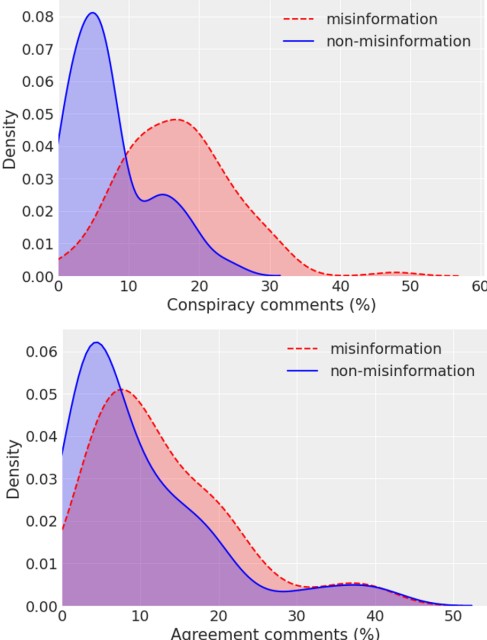

Figure 2: Probability densities of misinformation and factual videos regarding the percentage of conspiratorial comments (upper) agreement comments (lower).

| | LR | SVM | RF |
|---|---|---|---|
| **title** | 62.7 | **65.6** (l) | 64.4 |
| **conspiracy %** | 62.7 | **81.1** (r) | 72.2 |
| **comments** | 66.7 | **83.9** (r) | 82.8 |
| **title + conspiracy %** | 64.4 | 77.7 (s) | **82.2** |
| **comments + conspiracy %** | 73.3 | **89.4** (l) | 84.44 |
| **all** | 73.3 | **84.4** (l) | 82.7 |

Table 2: Classification accuracy for logistic regression, support vector machines, and random forest models for six feature settings. For the SVM, we applied three kernels: linear (l), sigmoid (s) and RBF (r). The kernel with the best accuracy appears in parenthesis.

## 3.3 Classification of YouTube Videos

The next step consisted of classifying the set of YouTube videos to detect misinformation. For this, we employed the percentage of conspiracy comments of each video as a feature. Additionally, we extracted content features from the videos' titles and from the raw first hundred comments per video (or all the comments for videos with fewer than 100 comments). For this, we preprocessed the titles and comments with tokenization, removal of stopwords, and the usage of the standard term frequency-inverse document (tf-idf) weighting for word frequencies to create a document term matrix, whose columns serve as input features. We selected six feature settings for our experiments: each of the set of features alone and the three possible combination between them . For each setting, we employed three classification models: logistic regression, support vector machine (SVM), and random forest. For the SVM models, we tried the linear, sigmoid, and RBF kernel. For both SVM and random forest, we performed a grid search to obtain the best hyperparameters. In each run, we performed 10-fold cross-validation and report the mean accuracy in Table 2. We observe that the SVM model has the highest accuracy for all the settings except for one. The conspiracy feature alone achieves an accuracy of 81.1. Using the tf-

idf comment features the accuracy is slightly better with 83.9. However, the conspiracy feature and comments combined achieve the highest accuracy of 89.4. We observe that the models with all the features combined have lower accuracy than the models omitting the title features. This may be due to overfitting and the title repeating information from the other two sets of features. Interestingly, the accuracy for the best model is still high (85.5%) when taking into consideration only videos with less than 100 comments. This implies that our methodology is appropriate for the early detection of misinformation videos.

## 3.4 Bayesian Modeling

To find the statistical validity of the conspiracy percentage feature, we turned to Bayesian modeling as it allows us to obtain the full posterior distribution of feature coefficients. We performed inference on three Bayesian logistic regression models using a Hamiltonian Monte Carlo solver. A simple model considered only the conspiracy percentage feature. A second model included this feature and the ten most relevant word features from the random forest model trained only on the title and conspiracy percentage. A third model included the conspiracy feature, and the top ten most relevant words from the linear SVM trained on the conspiracy feature and the first 100 comments. The first column of Table 3 and 4 shows the importance of each of the features in the random forest and linear SVM model, respectively. The two tables also show the statistics of the posterior probability distributions of the model coefficients: the mean, standard deviation, and the 1% and 99% quantiles. For the three models, the coefficients distribution converged (the $\widehat{R}$ diagnostic (Vehtari et al., 2019) was equal to one). We specifically selected logistic regression models for their interpretability. We observe that for the model based on the title word features, the

posterior distribution of the conspiracy percentage feature coefficient is the only one that does not include zero in its 98% highest posterior density interval (Table 3). Although this is not equivalent to traditional p-values, it conveys significance in a Bayesian setting. The model based on the 100 comments word features (Table 4), maintains the conspiracy feature as significant. However, also three coefficients from the word features avoid zero in their 98% interval. The model's coefficients are negative for *covid19* and *lab*, and positive for *god*.

Finally, we compare the three Bayesian models using the the WAIC information criteria, which estimates out-of-sample expectation and corrects for the effective number of parameters to avoid overfitting (Watanabe and Opper, 2010). Figure 3 shows the resulting deviance of the three models. We observe that the second model is slightly better than the simple model. However, the differences are included in the standard error of the title words feature model. This is not true for the simple model and the model including the comments features. In this case, the full model outperforms the model based only on the conspiracy feature. This indicates that there is important information in the videos' first hundred comments that is not explained by the conspiracy percentage feature on its own.

## 4  Discussion

We have leveraged large quantities of user comments to extract a simple feature that is effective in the prediction of misinformation videos. Given that the classifier is also accurate for videos with few comments, it can be used for online learning. For example, the user comments of videos containing *coronavirus* can be tracked and classified as they are posted. High levels of conspiracy comments could then indicate that the video includes misinformation claims. For this to work, it is not necessary a conspiracy classifier with perfect accuracy given that the percentage of conspiracy comments feature is based on an aggregated classifications. An improved classifier would be able to define a threshold that allows a balanced number of false positives and true negatives. The average percentage of conspiratorial comments would be maintained, irrespective of the wrong classifications. On the other hand, the accuracy of the video classifier is more critical. We found that using simple classifiers on the raw content of the videos' first 100 comments significantly improves the accuracy of misinformation video

|  | RF | mean | SD | 1% | 99% |
|---|---|---|---|---|---|
| conspiracy % | 19.2 | **28.25** | 4.8 | 18.19 | 39.94 |
| *coronavirus* | 2.95 | -7.45 | 3.4 | -15.57 | 0.01 |
| *covid19* | 2.81 | -5.17 | 2.4 | -11.08 | 0.10 |
| *china* | 1.42 | -4.28 | 3 | -11.23 | 2.63 |
| *man* | 1.24 | -6.04 | 2.8 | -12.25 | 0.52 |
| *bioweapon* | 1.24 | 4.81 | 5.5 | -6.40 | 19.32 |
| *conspiracy* | 1.1 | -4.24 | 3.7 | -13.96 | 3.72 |
| *new* | 1.03 | -5.13 | 5.4 | -18.93 | 6.39 |
| *update* | 0.87 | -0.15 | 2.5 | -6.57 | 5.69 |
| *cases* | 0.83 | -12.37 | 6.3 | -26.75 | 2.10 |
| *outbreak* | 0.72 | -1.25 | 2.9 | -8.31 | 5.66 |

Table 3: Top eleven features from the random forest model with the conspiracy and title as feature with the statistics of the coefficients' posterior probability distributions. The first column shows the percentage of feature importance.

|  | svm | mean | SD | 1% | 99% |
|---|---|---|---|---|---|
| conspiracy % | 2.82 | **34.96** | 6.2 | 20.56 | 50.09 |
| *virus* | 0.93 | -6.70 | 5.3 | -19.64 | 4.82 |
| *covid19* | 0.84 | **-28.8** | 10 | -54.33 | -6.20 |
| *god* | 0.75 | **19.29** | 7.6 | 3.39 | 37.54 |
| *allah* | 0.73 | -40.09 | 26 | -103.18 | 1.32 |
| *china* | 0.72 | -4.64 | 3.9 | -14.60 | 3.76 |
| *gates* | 0.69 | 3.39 | 16 | -32.39 | 42.94 |
| *amir* | 0.68 | -8.57 | 6.6 | -24.66 | 5.81 |
| *lab* | 0.68 | **-20.70** | 8.2 | -40.57 | -2.28 |
| *cases* | 0.66 | -22.41 | 14 | -57.26 | 8.48 |
| *trump* | 0.63 | 14.53 | 9.6 | -7.23 | 36.92 |

Table 4: Top eleven features from the SVM model with conspiracy and first 100 comments as features with the statistics of the coefficients' posterior probability distributions. The first column shows the SVM coefficients.

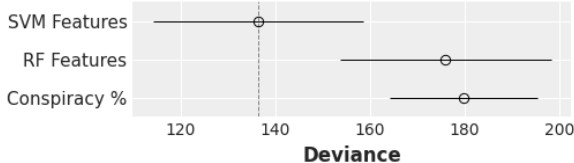

Figure 3: Deviance using WAIC as model selection metric. Black error bars represent the standard error.

detection from 82.2 to 89.4. However, in large-scale settings, it may be prohibitive to store the raw comments and continuously perform batch classification. In contrast, the conspiracy percentage feature only requires storing a conspiracy comment counter per video. Future research could leverage the video content to increase the classifier accuracy. The detection of misinformation on social media remains an open challenge, and further research is needed to understand how the COVID-19 misinfodemic spread to prevent future ones.

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
