# OpenReview forum: "NLP-based Feature Extraction for the Detection of COVID-19 Misinformation Videos on YouTube"
_aclweb.org/ACL/2020/Workshop/NLP-COVID — NLP-COVID-2020_

### Official Review · AnonReviewer1 · 2020-06-26
**Simple but effective method to detect COVID-19-related videos containing misinformation**

**Rating:** 7
**Confidence:** 3

**Review:**

This work proposes a simple but effective method to detect COVID-19 misinformation YouTube videos by using the percentage of conspiracy comments. While the authors did not examine video contents at all, the classification accuracy reached 82.2 by considering video title and the percentage of conspiracy comments only.

pros:
- The approach is simple but works well.
- The complete YouTube comments dataset and the Colab notebook file are publicly available.
- Great efforts were made to construct COVID-19-related misinformation video datasets.

cons:
- While I have some minor concerns/suggestions, none of them are serious flaws.

comments:
- Details in the labeling process should be added: For video labeling, did two authors independently label videos, and were all the labels agreed? For comments labeling, more explanations on the actual labeling process are required. For example, how many labels are collected per comment?
- Comparison with a bag-of-words model: As the authors already have all the text of comments, building a bag-of-words model based on the first 100 comments (videos with less than 100 comments were easy to predict) is not super hard but will make the author’s approach more concrete by adding another baseline. As there are several attempts to infer the political leaning of news articles based on their comments, bag-of-words models employing comments could be a solid baseline in this case as well.
- Agree & Conspiracy labels in Table 1: drawing 2x2 table might be helpful to understand the characteristics of Agree and Conspiracy as the authors mentioned ‘their different linguistic properties.’
- Crowdtangle’s historical data of public Facebook posts (Section 3.1): citations should be added.

---

> ### Author Response · Authors · 2020-06-29
> **Great suggestions! Now implemented**
>
> **Thank you very much for the great suggestions!** They allowed us to achieve better results and a more concrete approach. Here are our changes:
>
> >  Details in the labeling process should be added
>
> We added the necessary information about the labeling process. We had previously omitted it for spacing reasons. I agree that it is important to be transparent about labeling.
>
> > Comparison with a bag-of-words model
>
> Thank you for the great impulse! This changed the implications of our study. Using the first 100 comments has a good performance to classify COVID19 misinformation videos (83.9%). Adding the conspiracy percentage the new accuracy is 89.4%. We updated the discussion, abstract, and contributions to take into consideration these new insights.
> The results suggest that early detection of YouTube misinformation videos is possible, and could allow that videos like "plandemic" do not go viral and unnoticed for a long time.
> (Note: we used tf-idf instead of a bag of words model to tokenize the data. We tried both and tf-idf gave slightly better results. )
>
> > make the author’s approach more concrete by adding another baseline.
>
> This made us reflect on the conspiracy comment classifier. We did not provide baselines here either. We decided to change this and included two baseline models, one with LWIC and another one with Multinomial Naive Bayes. The difference between baseline and transformer models is better appreciated on a new precision-recall plot that we introduced for this classifier.
>
> > As there are several attempts to infer the political leaning of news articles based on their comments
>
> This is true! I added new citations on studies using user comments for prediction. Interestingly, Jiang and Wilson (2018) also used comments to detect misinformation. However, they only use linguistic signals and find no strong prediction value from user comments. (They did not separate between agree and misinformation labels for example)
>
> > Agree & Conspiracy labels in Table 1
>
> First, we eliminated Table 1 as the information was already in the text. We did not proceed to create the 2x2 table, but we performed $\chi^2$ tests on the bag of words of the agree and conspiracy comments. Less than 1% of the tokens appear in both datasets in a significantly similar manner. We hope this discussion helps to understand why the two types of comments are different linguistically.
> Our motivation for dividing the labels is that previous studies have used simply misinformation or believe in misinformation as a label and reported results on classifying such post. The problem with not dividing between comments such as "I agree with what you say" and "#pizzagate Justin Bieber is also part of the ring" is that the classifier will learn more to differentiate between agreement and real misinformation content.
>
> > Crowdtangle’s historical data of public Facebook posts
>
> We added the citation
>
> Additionally the code has been updated on GitHub.
>
> *With the above modifications, the paper is now 5 pages long... I know that for ACL short papers it is allowed to have 5 pages to address the reviewers' comments...I hope this is not a problem, otherwise, I would remove some plots to keep the 4 pages limit*

---

> > ### Comment · AnonReviewer1 · 2020-06-29
> > **Thanks for your R&R**
> >
> > Thanks for your quick r&r to address my concerns.
> >
> > I'm glad that the authors achieved better performance, believing that the study becomes more solid.
> >
> > I increased my rating. Thanks!
> >
> >
> > Regarding the paper length, please directly contact the workshop chairs.
> >
> > I'm not sure whether the resubmission's length violation is important or not in making a decision.

---

### Official Review · AnonReviewer4 · 2020-07-05
**An elegant approach and a great effort in data labelling but the authors need to address several questions**

**Rating:** 7
**Confidence:** 3

**Review:**

The study explores how user comments can be utilised to detect false information in Youtube videos related to COVID-19 origination. The authors did a great job reviewing and manually labelling a large number of videos and the corresponding comments. First, a model is created to detect user comments that convey agreement or amplify misinformation, referred to as conspiracy comments. Next, the percentage of conspiracy comments under each video is calculated and used along with video titles and raw comments to predict if a video contains conspiratorial content. This two-step approach allows the authors to avoid dealing with computationally-expensive video processing and efficiently detect COVID-19-related misinformation in Youtube videos.

Pros:
- A great effort in putting together the dataset of videos and comments;
- The paper is well-written and the ideas are clearly explained

Areas for improvement:
- I suggest clarifying in point 2 of the introductory section that other features (titles, raw comments) were used along with the percentage of conspiracy comments to detect COVID-19-related misinformation.
- There is a typo in “Pushshift Reddit API”
- It is said that conspiracy and agreement comments share 19.4% of their vocabulary and that only 1.95% of the vocabulary has more than four occurrences. Do you mean 1.95% of the shared vocabulary? Consider providing absolute numbers or restructuring the sentence to clarify the differences between the two collections.
- Did you use cross-validation to report the training F1 scores (Table 1)? Did you change the threshold after analysing the precision-recall curves (Figure 1)?
- I suggest changing non-misinformation to factual in the legend for consistency (Figure 2).
- Please clarify how the data was split for the development of the second model, there should be a hold-out set to test the performance of the final model. It would be great if the authors could provide a confusion matrix for the final model and analyse the false-positives/-negatives. Given the dataset is slightly imbalanced, reporting the F1 score instead of the accuracy would also be more appropriate.
- In Table 2, it would be good to provide the results for title + comments for benchmarking.
- Please comment on why you chose exactly 10 most relevant words for Bayesian modelling.

---

### Official Review · AnonReviewer3 · 2020-07-05
**A review of "NLP-based Feature Extraction for the Detection of COVID-19 Misinformation Videos on YouTube"**

**Rating:** 7
**Confidence:** 3

**Review:**

The paper describes a method for using YouTube video comments to classify video content, with the aim to separate videos into those describing Covid-19 conspiracy theories (misinformation) and those not (factual). An annotated video corpus is created by searching for videos on social media, downloading them from YouTube and annotating them, resulting in a dataset of 113 misinformation and 67 factual examples.

These videos have a total of 151,567 attached comments, which are further annotated as agreeing and/or conspiratorial, with manual annotation for the first 10% followed by automated propagation using a BERT-based neural network model. Finally, different classifiers are trained using the video description and comment label proportions to classify the video itself, and a feature analysis is performed to detect tokens important for this classification.

The paper is well written and the overall approach is well designed. The results are presented clearly and their significance is analysed thoroughly. The authors show that using attached comments as additional information can improve video classification, although since both the dataset and the method are new, it's not possible to compare or contrast the performance improvement with previous work. However, the authors have already published their dataset so further research and validation of the results is possible. Licence information (preferably Creative Commons for the dataset) should be added to the GitHub repository.

The paper follows a well-established approach of using contextual text as a proxy for audiovisual content classification. The current work in Covid-19 misinformation detection is a special case of this larger research area, and while the authors briefly mention this in citing Momeni et. al. (2013), the current work should be more extensively connected to this related work. For example, there appear to be at least two well known publications already from 2010-2011 that describe textual proxies for video classification [1,2] and it is commonly known that e.g. Google image search uses contextual text for visual content classification (https://support.google.com/webmasters/answer/114016) and has been doing so for a long time.

As a technical issue there may be something odd about how the parameters for the various classifiers are optimized. For example, the authors mention that "For both SVM and random forest, we performed a grid search to obtain the best hyperparameters. In each run, we performed 10-fold cross-validation and report the mean accuracy in Table 2." This sounds like hyper-parameters were first optimized on the full data on which the final results were later generated, which sounds concerning, and should be clarified.

Finally, it is not clear how much of this classification approach is already used by YouTube itself, which most likely uses contextual text in its video classification algorithms. The authors mention that "In the case of Covid-19 misinformation, a large share of conspiratorial contents remain online on YouTube and other platforms, influencing the public, despite content moderation practices", but then go on to state that "We decided not to use YouTube’s search function as previous studies found few conspiratorial content on the top results". Doesn't this indicate that YouTube can already detect these conspiratorial videos, reflected by their low ranking in the search results, even if they have not been outright removed?

As a very minor issue, there appears to be a typo in "the percentage of conspiracy comments feature is based on _an_ aggregated _classifications_."

[1] Huang, C., Fu, T., & Chen, H. (2010). Text‐based video content classification for online video‐sharing sites. Journal of the American Society for Information Science and Technology, 61(5), 891-906.

[2] Filippova, K., & Hall, K. B. (2011, July). Improved video categorization from text metadata and user comments. In Proceedings of the 34th international ACM SIGIR conference on Research and development in Information Retrieval (pp. 835-842).

---

### Decision · Program_Chairs · 2020-07-06

**Decision:**

Accept

**Comment:**

Thank you for your submission to the workshop.

Based on positive reviewer feedback, we are pleased to accept the paper for the workshop.

We look forward to your presentation on Thursday (5:30-9:30pm PDT) -- please plan on a 10-minute video presentation; pre-recorded is great.

Apologies for the late decision!